# Plant-Derived Bioactives in Oral Mucosal Lesions: A Key Emphasis to Curcumin, Lycopene, Chamomile, *Aloe vera*, Green Tea and Coffee Properties

**DOI:** 10.3390/biom9030106

**Published:** 2019-03-17

**Authors:** Bahare Salehi, Pia Lopez Jornet, Eduardo Pons-Fuster López, Daniela Calina, Mehdi Sharifi-Rad, Karina Ramírez-Alarcón, Katherine Forman, Marcos Fernández, Miquel Martorell, William N. Setzer, Natália Martins, Célia F. Rodrigues, Javad Sharifi-Rad

**Affiliations:** 1Student Research Committee, School of Medicine, Bam University of Medical Sciences, Bam 44340847, Iran; bahar.salehi007@gmail.com; 2Instituto Murciano de InvestigaciónBiosanitaria (IMIB-Arrixaca-UMU), Clínica Odontológica Universitaria Hospital Morales Meseguer Adv. Marques de los velez s/n, 30008 Murcia, Spain; majornet@ono.com; 3University of Murciaand, Clínica Odontológica Universitaria Hospital Morales Meseguer, Adv. Marques de los velez s/n, 30008 Murcia, Spain; edupfl5@hotmail.com; 4Department of Clinical Pharmacy, University of Medicine and Pharmacy of Craiova, 200349 Craiova, Romania; calinadaniela@gmail.com; 5Department of Medical Parasitology, Zabol University of Medical Sciences, Zabol 61663-335, Iran; 6Department of Nutrition and Dietetics, Faculty of Pharmacy, University of Concepcion, Concepcion 4070386, Chile; karramir@gmail.com (K.R.-A.); kforman@udec.cl (K.F.); 7Department of Pharmacy, Faculty of Pharmacy, University of Concepcion, Concepcion 4070386, Chile; marferna@udec.cl; 8Department of Chemistry, University of Alabama in Huntsville, Huntsville, AL 35899, USA; wsetzer@chemistry.uah.edu; 9Faculty of Medicine, University of Porto, Alameda Prof. Hernâni Monteiro, 4200-319 Porto, Portugal; 10Institute for Research and Innovation in Health (i3S), University of Porto, 4200-135 Porto, Portugal; 11LEPABE, Department of Chemical Engineering, Faculty of Engineering, University of Porto, Rua Dr. Roberto Frias, s/n, 4200-465 Porto, Portugal; c.fortunae@gmail.com; 12Food Safety Research Center (Salt), Semnan University of Medical Sciences, Semnan 35198-99951, Iran

**Keywords:** oral mucosal lesions, medicinal plants, plant extracts, phytochemicals, curcumin, lycopene, *Matricaria chamomilla*, *Aloe vera*

## Abstract

Oral mucosal lesions have many etiologies, including viral or bacterial infections, local trauma or irritation, systemic disorders, and even excessive alcohol and tobacco consumption. Folk knowledge on medicinal plants and phytochemicals in the treatment of oral mucosal lesions has gained special attention among the scientific community. Thus, this review aims to provide a brief overview on the traditional knowledge of plants in the treatment of oral mucosal lesions. This review was carried out consulting reports between 2008 and 2018 of PubMed (Medline), Web of Science, Embase, Scopus, Cochrane Database, Science Direct, and Google Scholar. The chosen keywords were plant, phytochemical, oral mucosa, leukoplakia, oral lichen planus and oral health. A special emphasis was given to certain plants (e.g., chamomile, *Aloe vera*, green tea, and *coffea*) and plant-derived bioactives (e.g., curcumin, lycopene) with anti-oral mucosal lesion activity. Finally, preclinical (in vitro and in vivo) and clinical studies examining both the safety and efficacy of medicinal plants and their derived phytochemicals were also carefully addressed.

## 1. Introduction

Oral mucosal lesions (OML) have been defined as any abnormal change in oral mucosa surface, appearing as pigmented, ulcerative, red and white features or any swelling or developmental defect variants [1]. Oral mucosal lesions have many predisposing factors, including infection (viruses, bacteria, parasites, fungi), thermal and/or physical, immune system deficiencies, neoplasia, systemic disease, trauma, as well as aging and chronic behaviors (e.g., betel nut, alcohol, and tobacco use) [2]. In 2003, OML was considered one of the main public health problems worldwide by the World Health Organization (WHO) [3]. In fact, it has been suggested that oral mucosa can reflect the patient’s general health [4]. 

More than 200 mucosal conditions have been identified [5]. The most important ones include leukoplakia, erythroplakia, oral lichen planus and lichenoid reactions, focal epithelial hyperplasia, exophytic neoplasia (oral fibroma and squamous papilloma/oral wart), herpes and aphthous lesions, tobacco-related lesions (white lesions associated with smokeless tobacco and nicotine stomatitis), non-specific ulcerations (defined as epithelium loss), candidiasis, traumatic ulcers and geographic tongue [6]. Oral mucosa is a highly responsive, diversified and dynamic environment that, although highly accessible, present true challenges for oral drug delivery [7]. Since steroids and other drugs that have been commonly used to treat different oral diseases present several side effects, scientists are searching for other methods with equivalent potency and little or no side effects [8]. 

Traditional knowledge on medicinal plants used for different health purposes has attracted much attention among the scientific community because of their effectiveness in the treatment of many diseases [9,10,11,12,13,14,15], including oral diseases [16]. There are many reports on the use of natural products and their derived bioactives, often called phytochemicals, in oral disease treatment [16,17,18]. It has been demonstrated that medicinal plants rich in several chemical constituents are highly effective in OML treatment. In this sense, based on the above-highlighted aspects, the present review aims to provide a close relationship between plant-derived bioactives and OML treatment efficacy. A special emphasis was also given to plant-derived bioactives with anti-oral mucosal lesion activity, with a special emphasis to curcumin, lycopene, chamomile, *Aloe vera*, green tea, and *Coffea*. Finally, preclinical (in vitro and in vivo) and clinical studies examining the safety and efficacy of these medicinal plants and their derived bioactives were also carefully examined. 

## 2. Research Methodology

Given the main focus of this work: o validate the traditional use of medicinal plants and their derived bioactives in OML treatment, the present review was carried out by consulting PubMed (Medline), Web of Science, Embase, Scopus, Cochrane Database, Science Direct, and Google Scholar (as a search engine) databases to retrieve the most updated articles on this topic. The following keywords were considered: plant, phytochemical, oral mucosa, leukoplakia, oral lichen planus and oral health. All articles were carefully analyzed by the authors to assess their strengths and weaknesses, and to select the more useful ones for the purpose of review, prioritizing articles published from 2008 to 2018.

## 3. Traditional Knowledge on Plants’ Use against Oral Mucosal Lesions

Medicinal plants have been used as folk remedies since ancient times and in many parts of the world. Specifically, their therapeutic applications in oral diseases include wound healing, anti-inflammatory, analgesic, antioxidant and antimicrobial benefits. Numerous reports have shown the effective potential of plant-derived bioactives in the treatment of OML, including recurrent aphthous stomatitis, mucositis caused by radiotherapy and chemotherapy, erosive leukoplakia and oral lichen planus. These phytopharmacological attributes have been documented in distinct traditional systems, such as Chinese, Ayurveda and Persian Medicine. 

### 3.1. Traditional Chinese Medicine

Chinese herbal medicine has been used for a very long time in the treatment of oral diseases. In general, each herbal medicine is a combination of several herbs. Thus, in the treatment of OML, it is possible to find different preparations of medicinal herb mixtures. Liuwei Dihuang consists of six ingredients extracted from *Rehmannia glutinosa*, *Cornus officinalis*, *Dioscorea opposita*, *Alisma orientalis*, *Poria cocos,* and *Paeonia suffruticosa*. The stomatitis-healing granule is a herbal mixture of *Asparagus cochinchinensis*, *Ophiopogon japonicus*, *Scrophularia ningpoensis*, *Lonicera japonica* and *Glycyrrhiza uralensis,* indicated for stomatitis [19]. Another preparation is Xianhuayin, a collutory (mouthwash), effective to treat oral, buccal mucosal premalignant lesions. Xianhuayin is a decoction consisting of *Phellodendrona murense*, *Amomum villosum*, *Sclerotium poriae*, *Helianthus,* and *Glycyrrhiza glabra*, where the main component in *G. glabra* is licorice zinc, which facilitates epithelial cell regeneration [20]. Nevertheless, it is also possible to find in Chinese medicine herbal preparations consisting of a single herb in the treatment of oral lesions. For instance, *Salvia miltiorrhiza* is a Chinese herbal medicine that has been used against tumor cell growth, inflammation and oxidation [21]. *Salvia miltiorrhiza* has been used in the treatment of scleroderma (an uncommon disorder characterized by thickening or hardening of the skin and fibrosis of the involved tissues) [22]. *Achyrantes bidentata* and *Achyrantes aspera* alcoholic extracts are often used in the treatment of aphtha (gargled) since they have demonstrated significant wound healing effects [23]. *Salvia officinalis, Matricaria chamomilla, A. vera,* and *Gentiana lutea* have also been used in traditional Chinese medicine as herbal formulae in the treatment of oral ulceration. They are often selected to be used against chemotherapy-induced mucositis, through the form of single infusions for gargling or topical application [24].

### 3.2. Ayurveda, Indian Traditional System of Medicine

Indian herbal medicine also has a thousand years of rich traditional knowledge in wound treatment and management. Among the wide variety of plants used in wounds management, it is possible to differentiate those intended for specific oral lesions treatment. *Aloe vera* is an indigenous medicinal plant found throughout India [25]. This plant is effective in the treatment of oral aphthous ulceration [26], but has been reported as having many other properties, such as antiulcer, antiseptic, antibacterial, anti-inflammatory, antioxidant, and wound healing properties [8,27]. *Centella asiatica* is distributed throughout the plains of India [28]. This plant is effective in the treatment of mouth ulcers. It has a remarkable effect on wounds healing and promotes connective tissue growth [29]. The wound healing effect has been attributed to several mechanisms, including antioxidant activity, collagen synthesis, and angiogenesis promotion [30]. *Curcuma longa* is another very useful and old spice used in Ayurvedic medicine. *C. longa* extracts can be used in the treatment of oral cavity lesions [31]. One of the most important components of *C. longa* is curcumin, a potent antioxidant. Also, *C. longa* has prominent anti-inflammatory, antibacterial and wound-healing effects [32]. *Emblica officinalis* has an interesting antioxidant and astringent property, and it has been demonstrated to be effective in the treatment of aphthous stomatitis and other types of mouth ulcers [33]. *Tinospora cordifolia* has anti-inflammatory, antioxidant and immunomodulatory effects. This plant can reduce mucositis severity in radiotherapy patients [34]. *Jasminum grandiflorum* leaves are widely used in the treatment of ulcerative stomatitis and oral wounds, closely related to its antioxidant properties [32].

### 3.3. Traditional Persian Medicine

Wounds treatment is also an important feature in traditional Persian Medicine, where some plants can be effectively used to treat diseases of the oral cavity. *Lawsonia alba* is used as a rinsing decoction in aphtha treatment. Its wound healing effect is mediated through its anti-inflammatory, antioxidant and antibacterial activities [35]. *Punica granatum* is also used as a rinsing decoction with vinegar to treat aphtha through its anti-inflammatory, antioxidant and antibacterial activities [36,37]. Finally, *Vitis vinifera* seed extract also displays anti-inflammatory and antioxidant effects [38].

## 4. Curcumin, Lycopene, Chamomile, *Aloe vera*, Green Tea and Coffee in Oral Mucosal Lesions: A Brief Overview

The mouth is considered the human organism gateway, with many functions including nutrition, communication (speech) and facial expression [39]. In recent decades, there has been increasing interest in assessing the therapeutic effects of plants for the treatment of oral mucosal lesions [40,41]. Plant-derived preparations interact through different biochemical pathways, which culminates in distinct biological potentialities, including antioxidant, analgesic, anti-inflammatory, antifungal, antiseptic and anticarcinogenic effects (Figure 1). 

Curcumin is a polyphenol commonly known as turmeric. Its main active components include flavonoids and volatile compounds such as tumerone, atlantone, and zingiberone. Recent scientific evidence has shown that turmeric, and particularly curcumin, exhibit powerful anti-inflammatory effects in a wide variety of target systems [42,43,44]. Also, remarkable antioxidant, anti-tumor, analgesic, immunostimulant, antiviral, antibacterial and antifungal effects have been attributed to this molecule, as well as a great capacity to fight against diseases, such as diabetes, asthma, allergies, neurodegenerative, arthritis, atherosclerosis, OML, and even cancer.

Lycopene is a carotenoid and natural liposoluble pigment responsible for the red and orange color of some fruits and vegetables. An important factor affecting lycopene bioavailability is its synergy with other antioxidants, as occurs, for example, with vitamins C and E. Carotenoids’ toxicity stated in some observational and interventional studies is closely related to the doses used and its corresponding interactions. It should be noted that most of these studies have used rodents, which absorb carotenoids less efficiently than humans. Also, high concentrations of one carotenoid may interfere with the bioavailability of others, leading to imbalance, as occurs between β-carotene and lycopene [45].

Chamomile (*Matricaria recutita* L., syn. *Matricaria chamomilla* L.) is a medicinal plant belonging to the Asteraceae family and contains flavonoids, coumarins and essential oils with antiseptic, carminative, sedative and protector effect against mucosal ulcers [46,47,48,49,50].

The *Aloe vera* species contains over 200 biologically active substances, such us anthraquinones (barbaloin, isobarbaloin, anthranols, aloetic acid), hydrosoluble and liposoluble vitamins, minerals, enzymes, polysaccharides, phenolic compounds, and organic acids, with antibacterial, antimicrobial, anti-inflammatory and immuno-modulatory effects [51,52,53,54,55,56]. Lignin is very abundant in *A. Vera* pulp and easily penetrates into epithelial tissues, carrying other compounds such as saponins (glucosides with a cleansing and antiseptic activity), chrysophanic acid and emodin derivatives used in the treatment of psoriasis outbreaks and skin mycoses [57].

Tea is a product made from *Camellia sinensis* leaf and bud, and it is the most widely consumed drink in the world after water, being second only to water in popularity as a beverage [58]. This is an important source of polyphenols with renowned antioxidant, anti-inflammatory and bactericidal effects [59]. 

## 5. In Vitro and In Vivo Anti-Oral Mucosal Lesions Effects

### 5.1. Curcumin: Effects on Oral Mucosal Carcinogenesis and Anti-Inflammatory Properties

#### 5.1.1. Effects on Oral Mucosal Carcinogenesis

Curcumin modulates key pathways involved in inflammation and carcinogenesis [42,60]. In vivo studies in rats fed with curcumin at 0.5 g/kg during initiation and post-initiation phases showed a 91% reduction in 4-nitroquinoline-1-oxide-induced tongue carcinoma frequency [61]. Furthermore, the incidence of potentially malignant *N*-nitrosomethylbenzylamine-induced oral lesions decreased as a result of curcumin administration during the initiation and post-initiation phases [62]. In another study, where curcumin was administered alone and in combination with green tea (*C. sinensis*), it was found that it exerts inhibitory effects against oral carcinogenesis in hamsters, suppressing cell proliferation, inducing apoptosis and inhibiting angiogenesis [63,64]. Manoharan et al. [65] induced oral squamous cell carcinoma (OSCC) in hamsters through using 7,12-dimethylbenz[a]anthracene (DMBA). Curcumin and piperine administered orally in hamsters with DMBA-induced OSCC hampered oral tumor formation, probably due to their antioxidant effects. Lin et al. [66] observed that curcumin significantly inhibited human OSCC cell (SAS cell line) proliferation and growth in mice subcutaneously inoculated. Indeed, curcumin cytotoxic effect targeted the G2/M phase of the cell cycle. On the other hand, Lee et al. [67] also found that copper supplements significantly improved curcumin inhibitory effect against oral cancer cell migration and viability. Recent studies have also investigated the effects of curcumin radiosensitization in OSCC cell lines and in head and neck squamous carcinoma cells (HNSCC) [68,69,70]. It was demonstrated that the combination of curcumin and irradiation had additive effects [60]. In another study, using SCC-1 cell lines, curcumin decreased cyclooxygenase-2 (COX-2) expression and inhibited epidermal growth factor receptor (EGFR) phosphorylation [71]. Lopez-Jornet et al. [68] found that curcumin administration in OSCC cells (PE/CA-PJ15), exposed to different irradiation doses (1.0, 2.5 and 5.0 Gy), led to an increase in cytotoxic activity, acting in OSCC cells. Tuttle et al. [70] examined curcumin-induced radiosensitization in HNSCC cell lines at different stages of human papillomavirus (HPV) infection, finding that curcumin increased the radiation response in HPV-resistant (-) cell lines, but had no effects in HPV (+) cells. In general, curcumin biological safety, combined with its low cost and efficacy, as well as thousands of years of experimentation, justifies it’s being known as the Indian solid gold.

One of the criticisms of curcumin research is that most evidence on its therapeutic potential is based on in vitro studies. Therefore, the ideal dose to treat any disease is still unclear. In addition, there are some additional components in turmeric that may have beneficial effects, alone or in combination with curcumin, and some studies suggest that agents such as piperine may increase the bioavailability of curcumin. Well-managed clinical trials are thus required to determine curcumin potential in the prevention and treatment of both diseases [70,72,73,74]. In light of its potential properties, curcumin can be very useful in dentistry and play an important role in the treatment of periodontal disease, OML, and even oral cancer.

#### 5.1.2. Anti-Inflammatory Properties

Curcumin’s pharmacological safety, combined with its anti-inflammatory action makes it an effective therapeutic and preventive agent for treating inflammatory disorders [75]. Among the existing data, it has been found that curcumin modulates cell activity of various growth factors, cytokines, and transcription factors, often involved in inflammatory processes [42,60]. The in vitro inhibitory effects of curcumin on pro-inflammatory cytokines production in human monocytes and macrophages has been studied with lipopolysaccharide (LPS). Curcumin showed the ability to inhibit interleukin 8 (IL-8) production, monocyte chemoattractant protein-1 (MCP-1), interleukin 1 beta (IL-1β) and tumor necrosis factor-alpha (TNF-α). Also, it has been suggested that curcumin’s inhibition of cytokine production could be due to the inhibition of different signaling pathways activation, such as the protein kinase C (PKC) pathway (Figure 2 and Table 1). 

Curcumin anti-inflammatory effects have also been associated with its chemical structure, as it possesses two phenyl and methoxy groups in the ortho position, which have been shown to have a great ability to inhibit nuclear factor kappa B (NF-κB) and so halt TNF-α and IL-6 production [42,60]. It is likely that curcumin anti-inflammatory effect is mediated through its ability to inhibit COX-2, lipoxygenase (LOX) and inducible nitric oxide synthase (iNOS), important enzymes that mediate inflammatory processes [44]. IInadequate positive regulation of both COX-2 and/or iNOS has been linked with the physiopathology of certain types of cancer in humans, as well as with inflammatory disorders [85]. The fact that inflammation is closely related to tumor promotion indicates that curcumin, with its powerful anti-inflammatory effects, may exert chemopreventive effects on carcinogenesis [86]. Indeed, curcumin modulates key pathways involved in inflammation and carcinogenesis [42,60]. It has been shown that curcumin protects against the skin, oral, intestinal and colon cancer, and also suppresses angiogenesis and metastasis in a wide variety of models [87]. It also inhibits cancerous cell proliferation, detaining cells in different phases of the cell cycle, and inducing apoptosis [88]. Furthermore, curcumin has a promising capacity to inhibit carcinogenic activation using specific cytochrome P450 isoenzymes suppression, as well as by inducing the activity or expression of carcinogen-detoxifying phase II enzymes [89]. Further pharmacodynamic studies are needed to supply sufficient data in support of clinical assessment of curcumin’s chemopreventive potential.

### 5.2. Lycopene: Antioxidant and Chemopreventive Properties

#### 5.2.1. Antioxidant Properties

Lycopene has shown beneficial effects in the treatment of OML based on its antioxidant properties, inhibition of cancer cell proliferation and interference with growth factor stimulation, phase II enzyme induction, transcription regulation and gap junction restoration [90,91,92,93,94,95,96,97]. The mechanisms underlying lycopene’s inhibitory effects on carcinogenesis seems to involve reactive oxygen species (ROS) elimination, positive regulation of detoxification systems, interference in cell proliferation, induction of Gap-junction communication, or inhibition of cell cycle progression. Also, it has been reported that lycopene increases p53 protein levels in cancer cells [98]. Lycopene appears to be a promising antioxidant as a treatment modality for oral leukoplakia; it can protect cells against damage and plays a protective role against oral dysplasia progression through inhibiting tumor cell proliferation. Indeed, initial reports regarding lycopene efficacy on human oral cancer cells describe significant therapeutic effects [99,100,101,102]. In addition, and given its antioxidant potential, it has been hypothesized that carotenoid ingestion can reduce the risk of head and neck cancer (HNC). This agrees with the information provided by several major studies that analyzed the overall carotenoids impact in HNC. Participants with low carotenoid intake, who smoke heavily, or who drink alcohol appeared to present higher HNC risk [103,104]. 

#### 5.2.2. Chemopreventive Properties

Bhuvaneswari et al. [105] studied the chemopreventive properties of lycopene and tomato paste in an oral carcinogenesis hamster model. Both significantly reduced the incidence of oral mucosa tumors modulated lipid peroxidation and enhanced reduced glutathione (GSH) and GSH-dependent enzymes in the cheek pouch, liver, and erythrocytes [106,107]. These results suggest that lycopene exerts its anticarcinogenic effects through different mechanisms [102]. Cheng et al. [108] investigated the chemopreventive effect of lycopene and other carotenoids in a betel quid extract-induced hamster oral cancer model. Lycopene and mixed carotenoids prevented carcinoma apparition, while in dysplastic lesions, which appeared in all groups, the expression of proliferating cell nuclear antigen was less in the lycopene group when compared with controls [108]. 

### 5.3. Chamomile: Antioxidant, Antimicrobial, Oral Mucosal Protector and Anti-Carcinogenic Effects

#### 5.3.1. Antioxidant Properties

Bisabolol and chamazulene possess potent antioxidant effects [96]. One study conducted in animals assessed the protective effects of chamomile extracts against ROS. Chamomile plant extract was able to prevent reactive chemical species production and to block lipid peroxidation through different chemical processes. Furthermore, chamomile-extracted chamazulene inhibited Fe^2+^/ascorbate-induced lipid peroxidation and dimethyl sulfoxide (DMSO) autoxidation [46,47,109,110,111].

#### 5.3.2. Antimicrobial Properties

Chamomile essential oils have also exhibited antimicrobial effects against certain bacteria, fungi, and viruses [112]. The antibacterial effect of chamomile extract fractions was assessed against two Gram-negative bacteria. Results obtained confirmed its antibacterial effect, conferred by its main components, coumarin, flavonoids, phenolic acids and fatty acids [113]. Chamomile has moderate in vitro antioxidant and significant antiplatelet activity [48,49,50,109,114,115,116]. Studies in animal models indicate that it has a potent anti-inflammatory activity, some antimutagenic and hypolipidemic effects.

#### 5.3.3. Oral Mucosal Protector

Chamomile has been investigated as an oral mucosa protector and can also be effective for treating or preventing mouth ulcers triggered by chemo or radiotherapy. However, topical chamomile application through a form of a mouthwash is contradictory, although there are possible reasons to explain the distinct results obtained such as the composition and implementation characteristics of the drug used in each study, the type of chamomile and the way through which the plant was handled or processed [117,118,119,120]. Also, the clinical efficacy of topical administration depends on factors inherent to medication (chemical structure, concentration, vehicle used, presence of other substances), implementation (applications number andway, frequency and duration) and of the individual subject (nature, location and lesion degree, individuals age and gender) [118]. Moreover, the type of chamomile used and the way through which it was handled can influence the results obtained. For example, it was shown that qualitative and quantitative differences in chamomile essential oil are not affected by cultivation conditions (i.e., fertilization, irrigation, pesticides use), but can significantly vary between the distinct regions where chamomile grows, between cultivated and wild chamomile, and under different processing conditions [112]. 

#### 5.3.4. Anti-Carcinogenic Properties

There is a growing interest in the use of apigenin, the main chemical compound of *Chamomilla* as a health-promoting agent because it has recently been recognized as a chemoprotector, due to its powerful antioxidant and anti-inflammatory effects. For instance, it has been shown that apigenin exerts antimutagenic and antiviral effects [46,47,109,110,121,122,123]. Apigenin impairs OSCC cells due to its cytotoxic effects and its ability to act as cell cycle modulator [124]. Further it can inhibit hypoxia-induced stem cell marker expression [125]. Apigenin’s antitumor effects are due to the fact that apigenin regulates all carcinogenesis stages. In the initiation phase, apigenin protects the DNA of different cells from damage induced by genotoxic compounds, avoiding mutations that could provoke tumor development. In this way, it can promote metal chelation, regulate enzyme activities related to carcinogens metabolism, such as cytochrome P450, eliminate free radicals, and stimulate phase II detoxifying enzymes [122]. In the promotion phase, apigenin inhibits or slows cell division through mediating the cell-cycle regulation and inducing apoptosis. It acts as a cell-cycle regulator in many cell lines. It also reduces the levels of various cyclins (A, D1, 2 and B1), deactivates cyclin-dependent kinase (CDK), and positively regulates CDK inhibitors. Moreover, it has the ability to stabilize p53 protein, responsible for activating p21/WAF1 protein and for inducing Rb protein dephosphorylation, which in turn avoids cyclin D and E expression related to cell cycle. Its pro-apoptotic activity is due to the fact that it increases suppressant p53 protein expression, which frequently evolves cancer, reduces Bcl-2 and Bcl-xL protein expression (anti-apoptotic effect), increases Bax and Bak expression (pro-apoptotic effect), increases the exit flow of cytochrome c with the consequent caspase-9 and caspase-3 activation, activates caspase cascade, and inhibits type II DNA topoisomerase [122]. Finally, in the last phase of carcinogenesis, progression phase, it has been shown that apigenin inhibits angiogenesis and metastasis processes. The effects of apigenin would appear to be mediated mainly through hypoxia-inducible factor 1-alpha expression, COX-2 expression and nitric oxide-2 synthesis suppression, and vascular endothelial growth factor (VEGF) and lipoxygenase reduction [124].

### 5.4. Aloe vera: Oral Lesions Healing and Tissue Regeneration, Immunomodulatory, Anti-Inflammatory, Antioxidant, Antibacterial, Antifungal, Antiviral, and Anti-Carcinogenic Effects

*Aloe vera*’s complex and heterogeneous composition gives rise to numerous pharmacological actions that have been confirmed in diverse research [52,53,54,55,56,126,127,128]. These include healing and tissue regeneration, anti-inflammatory, analgesic, bacteriostatic and bactericide, antioxidant and anti-diabetic effects. Below is presented a brief outline of these effects, also summarized in Table 2.

#### 5.4.1. Oral Lesions Healing and Tissue Regeneration

*Aloe vera* has a great capacity to increase cell proliferation and help in the healing process of oral lesions, rapid and effectively; the same applies to all kinds of skin lesions [133]. These effects have been confirmed in animal studies [126,127,128,129,134,135,136]. This direct effect on healing process manifests as an increase in the rate of lesion area contraction and is directly attributed to the presence of mannose-6-phosphate [126]. In fibroblasts, *Aloe* polysaccharides promote both fibroblast proliferation and hyaluronic acid and hydroxyproline production, which play an important role in extracellular matrix remodeling during lesions healing [136]. In vitro studies have found that the *A. vera* glycoprotein fraction accelerates healing and cell migration in human keratinocytes [129]. The encapsulation of *A. Vera* liposomes helps in collagen proliferation and synthesis in human skin cell lines [137]. Some of the polysaccharides isolated from *A. vera* include acemanans, which have powerful cell regenerating properties and are able to promote gingival fibroblast proliferation, collagen type I production, and increases VEGF and keratinocyte growth factor I (KGF-I), effects that have been related to oral lesion healing in rats [135]. It acts on periodontal ligament cell proliferation, positive growth factor regulation/differentiation, and collagen type I and alkaline phosphate activity in primary human periodontal ligament cells [134].

#### 5.4.2. Immunomodulatory Potential

*Aloe vera* gel has strong immunomodulatory effects using macrophage cell activation, that generates nitric oxide, secretes cytokines (like TNF-α, IL-1, IL-6, and interferon alpha (INF-α)), and presents cell surface markers [52,53,54,55,56].

#### 5.4.3. Anti-Inflammatory Potential

Recent studies in *A. vera’s* anti-inflammatory mechanisms of action suggest that they are probably produced by an inhibitory effect on arachidonic acid pathway production, by means of cyclooxygenase [53,54,55,56]. 

#### 5.4.4. Antioxidant Potential

It has been shown that *A. vera* has a pronounced antioxidant content regardless of whether it is administered orally or locally applied to wounds. Main antioxidants that are able to protect against oxidative stress and trigger cell death [51,52,53,54,55,56,126], include phenols, such as aloin, barbaloin and isobarbaloin, among others, of which the most significant are vitamins A, C, and E (with protective activity on skin, mucosa, and lipid components), and minerals, such as selenium, zinc or copper (that help to form hydrosoluble complexes or that participate as enzymatic cofactors) [138]. 

#### 5.4.5. Antibacterial, Antifungal, and Antiviral Potential

In vitro studies have shown that *A. vera* display antimicrobial effects against isolated gram-negative and gram-positive bacteria [55,139]. Other studies have stated its powerful antifungal activity against different *Candida* species, especially *Candida parapsilosis, Candida krusei* and *Candida albicans* [139,140]. Research has also suggested that *A. vera* has antiviral action preventing virus absorption, fixation and entry into host cells. It has also been shown that *A. vera* gel has an antiviral effect against herpes simplex virus type 2 strains [141].

#### 5.4.6. Anticarcinogenic Potential

*Aloe vera,* mainly due to its anthraquinone compound, aloe-emodin, is a good agent in cancer therapy, since it induces cell apoptosis, among other mechanisms [52,53,54]. Aloin treatment, another anthraquinone present in *A. vera*, has shown to inhibit angiogenesis and tumor growth. In vitro studies have found that aloe-emodin halts the cell cycle of nasopharyngeal carcinoma through metalloprotein expression and p38 pathway and NF-κB induction. This also induces apoptosis through caspase activation [142,143]. In an in vitro study using human oral cancer cells, aloe-emodin increased alkaline phosphate activity and induced G2/M cell cycle arrest, which indicates that it could help in oral cancer treatment [144]. Xiao et al. [145] investigating the oral cancer cell line found that aloe-emodin compound reduced cell proliferation and migration in human oral cancer through protein kinase C inhibition. Thus, given its beneficial effects, in particular the mechanism of action by which *Aloe* exerts its protective and/or preventive effects, this plant, it’s extracts and even compounds may be conceived as having a great potential to be used for pharmaceutical, cosmetic and nutritional purposes, helping to reduce the risks and secondary effects of many diseases. 

## 6. Clinical Effects of Phytochemicals in Oral Mucosal Lesions in Humans

### 6.1. Curcumin

Curcumin exerts beneficial effects in patients with oral lichen planus and leukoplakia [146,147]. Curcumin caplets (900 mg + 80 mg desmethoxycurcumin + 20 mg bisdesmethoxycurcumin) increased antioxidant levels, such as vitamins C and E, in saliva and serum, and prevented lipid peroxidation and DNA damage [146]. This study was conducted in 100 patients with oral leukoplakia, lichen planus, oral submucous fibrosis and in healthy subjects rankging from 17 to 50 years. The diverse parameters, such as pain, measured by a visual analog scale, and lesion size in all illnesses studied were significantly improved with treatment. The mouth opening in oral submucous fibrosis patients was also remedied (or relieved) (Table 3).

Curcuminoid’s efficacy, assessed using high doses (about 6 g/day, divided into three doses) for 12 days in a randomized, double-blind, placebo-controlled study, has been related to a decrease in oral lichen planus signs and symptoms in participants older than 21 years [147]. Moreover, at high doses, a greater reduction in symptoms was measured, using the numerical rating scale (NRS) and total modified oral mucositis Index (MOMI) score; both quantification scales have been validated for the measurement of oral lichen planus-associated symptoms [148]. Also, a significant reduction in signs was stated as measured by percentage change in erythema and in ulceration level as compared with the placebo group [148]. On the other hand, studies using lower doses (2 g/day of a standardized turmeric extract containing 95% curcuminoids, divided into two doses) for seven weeks revealed to be ineffective [147].

In young people, the use of curcumin to treat oral lichen planus generated positive results [149]. At doses tested (1 g for two weeks, and, then, tapered to 500 mg for two weeks), clinical signs and symptoms disappeared with no adverse effects [149]. On the other hand, different formulations containing curcumin together with other bioactive matrices, such as *Bidens pilosa* extract, have been developed for the prevention and treatment of chemoradiotherapy-induced OML, also testing the safety of these compounds [150]. In this study, 20 healthy adults were randomized into two groups, who received different formulations, 10 mg/mL or 20 mg/mL of curcuminoids extract plus 20% or 40% *v*/*v* of *B. pilosa* extract, for 10 days, three times a day. None of these formulations increased pro-inflammatory cytokines levels, such as IL-1, IL-6 or TNF-α, nitric oxide concentration or induced micronucleus frequencies changes, a marker of genotoxicity in oral mucosa cells [151].

### 6.2. Lycopene

The efficacy of lycopene in the treatment of oral diseases has also been investigated in various studies [76,152,153,154]. In a study conducted in 20 patients with moderate periodontitis or moderate gingivitis, two treatment groups were designed, one received 4 mg lycopene/day for two weeks with oral prophylaxis, and controls received only oral prophylaxis [152]. 

Lycopene exerted it’s effect as an adjunct to oral prophylaxis, based in full-mouth scaling and root planing, in patients with moderate periodontal disease; thus, the authors concluded that the inhibition of ROS production was the mechanism involved [152]. In another study, conducted in 20 patients with gingivitis, 10 ingested 8 mg lycopene for two weeks, and the control group consumed a placebo preparation [76]. 

Lycopene significantly reduced gingivitis, bleeding index and non-invasive measures of plaque [76]. Also, a case-control study performed in Japan, including 9536 subjects over 40 years, found that among males with leukoplakia, mean serum lycopene and β-carotene levels were significantly lower than those of controls [153]. Moreover, lower lycopene levels were found in patients with atrophic/erosive oral lichen planus [154].

### 6.3. Green Tea and Coffee

Green tea is a rich source of flavonoids, such as catechins that evidenced bactericidal effects in a clinical study of periodontal disease [155]. In fact, periodontal status was improved using a combination of mechanical treatment and local application of green tea, rich in catechins, for eight weeks, through slow-release hydroxyl-ethyl-cellulose oral strips [155]. Another clinical study, performed in 30 patients with chronic periodontitis and ranging in age from 18 to 60, demonstrated the beneficial effects of green tea in periodontitis severity indicators [156]. The clinical measurement of periodontal disease included probing depth (PD), gingival index (GI) and clinical attachment level (CAL). In this study, patients were advised to brush twice daily (2–5 min) using a green tea dentifrice containing 60% to 90% epigallocatechin, which was able to improve PD, GI, and CAL in these patients [156]. Moreover, green tea treatment significantly increased antioxidant activity through increasing glutathione-*S*-transferase [156]. 

On the other hand, caffeine is the most routinely ingested bioactive substance throughout the world. It is a natural alkaloid present in coffee, tea and cocoa, and with different properties. Caffeine has antioxidant and anti-inflammatory effects [165], and its protective effects against oxidative DNA damage in rats has also been reported [166]. An analysis of pooled case-control studies from nine HNC studies found an inverse association between coffee drinking and the risk of cancer of the oral cavity and pharynx [157]. Information was collected as cups of caffeinated or decaffeinated coffee, and cups of tea per day [157]. Also, a double-blinded randomized clinical trial of a total of 75 participants with oral mucositis investigated the therapeutic effects of coffee, honey and the combination of both on mucositis lesions [158]. The tested doses were 300 g of honey and 20 g of instant coffee every 3 h for one week. In this study, every treatment reduced lesion severity, which suggests that oral mucositis can be successfully treated, particularly using a combination of honey and coffee [158]. 

Daneshyar et al. showed that the application of green tea varnish could inhibits root caries, which is a common and debilitating condition, mainly in the elderly population, that can lead to tooth loss. Indeed, the authors found that following green tea application, anti-cariostatic effects were prominent, with remarkable differences in carious lesions, when applied every 24 or 48 h over 21 days [167]. In a systematic review related to leukoplakia and oral cancer prevention, it was reported that some authors demonstrated benefits in the treatments with green tea, when compared with placebo [168]. On the other hand, Ghorbani et al. treated 22 patients with *Candida* species-induced denture stomatitis using green tea mouthwash 0.5% and nystatin suspension 100,000 U/mL. The authors found that green tea displays a comparable anti-*Candida* species activity regarding nystatin, making it a possible alternative treatment [169]. Other tactics have also been studied. The combination of EGCG with miconazole, fluconazole or amphotericin B has been revealed to have a synergistic effect against planktonic and biofilm cells of *Candida* species. As in the main findings, it was stated that this combination could lower the dosages of antifungal drugs needed to treat fungal infections, inhibiting adverse effects and the emergence of resistant strains [170,171]. Another report described an evaluation of the effects of tea and coffee intakes on oral OSCC stratified by milk intake. Yan et al. explained that tea intake was significantly correlated with a decreased OSCC risk. Also, there was a link between starting to drink tea at age 25 or later, drinking green tea and oolong tea, moderate tea concentration, and water temperature and a decrease in OSCC risk. A multiplicative, but not additive interaction, was found amongst tea drinking and milk intake, but no connection was detected in the case of coffee intake and OSCC (with or without milk intake) [172]. 

Regarding, specifically, coffee, Song et al. [173] evaluated the link between coffee intake and tooth loss in Korean adults. The results showed that the prevalence of having less than 20 remaining teeth was 69% higher in groups with daily coffee intake than those with coffee intake of less than once a month. These findings demonstrate that daily coffee consumption is closely associated with tooth loss in Korean adults. However, this outcome may be possibly related to sugar intake (carbonated additives) added to beverages, as the same authors have clearly demonstrated [174].

There are conflicting results on the influence of coffee in oral and pharyngeal cancer risk. Miranda et al. performed a systemic review and meta-analysis on this matter and concluded that the results present an inverse association between high coffee consumption and the risk of oral and pharyngeal cancers, indicating that coffee may have a protective role against these cancers. However, the authors indicate that further larger prospective observational cohort studies are needed to address any effect of other possible co-factors [175]. Other authors reached similar conclusions [176]. 

### 6.4. Chamomile

As already mentioned, chamomile has several bioactive effects, including upon oral health. Chamomile has antiseptic and anti-inflammatory properties, which together with its moderate antioxidant and antimicrobial activities confer protection against ulcers [109,112]. Some studies have been evaluated the chamomile potential as a mucosal protective agent, but the obtained results are controversial [177]. In a clinical case of methotrexate-induced oral mucositis, conducted in an elderly patient with rheumatoid arthritis, the beneficial effect of wild chamomile mouthwashes 4 times daily was demonstrated [159]. Chamomile was administered adding 8 g of the dried flowers into 1000 mL of boiling water, covering and infusing for 15 min. The suggested amount and duration of each mouthwash was 20 mL of the preparation for 1–2 min [159]. 

A randomized pilot study was performed in patients with cancer receiving 5-fluorouracil and leucovorin to investigate the chamomile infusion (2.5%) effect in oral mucositis prevention and reduction of intensity at day 8, 15, and 22 after the first day of chemotherapy [160]. The parameters analyzed were mouth pain, by a numerical 10-point scale [178], mucositis level, measured by the WHO assessment scale reference [179], and the presence of ulceration. Chamomile group showed lower mouth pain score and fewer ulcerations than the control group, besides never having developed high levels of mucositis.

### 6.5. Aloe Vera

The role of *A. vera* in oral illness has also been widely studied in the context of oral lichen planus, radiation-induced mucositis, burning mouth syndrome, xerostomia, recurrent aphthous ulcers, among others [8]. *A. vera* has recently gained a huge importance in oral mucositis prevention in patients receiving systemic anticancer therapies [180]. But, although the current *A. vera* benefits in reducing disease severity, the evidence is not conclusive. A double-blind, prospective, randomized trial was developed to study the efficacy of *A. vera* oral gel on radiation-induced mucositis in HNC patients [161]. This study assessed mucositis severity, head-and-neck radiotherapy tolerance, pain level, quality-of-life perception, analgesic therapy, oral infections, among other parameters. No significant differences were observed [161]. On the other hand, it has been suggested that *A. vera* mouthwash may prevent radiation-induced mucositis by promoting wound healing and reducing inflammation. Furthermore, *A. vera* antifungal and immunomodulatory effects may reduce oral candidiasis severity in patients with head and neck radiotherapy [162]. 

Topical *A. vera* application has been associated with the treatment of oral lichen planus [163]. In 46 patients with oral lichen planus, *A. vera* mouthwash reduced the visual analogue score developed by Thongprasom, a scale that measured pain as well as the presence and extent of the white stretch, erythema, and atrophy after treatment. Also, *A. vera* proved to be as useful as triamcinolone acetonide 0.1% [163]. In a randomized, double-blind, placebo-controlled trial conducted in 54 patients with oral lichen planus for eight weeks, the efficacy of *A. vera* gel (70% *A. vera* mucilage, sorbitol, potassium sorbate, sodium metabisulphite, and hydroxyethylcellulose) was compared to placebo in the topical management of oral lichen planus [164]. *A. vera* gel treated group demonstrated clinical and symptoms improvement [164].

## 7. Conclusions

This review brings knowledge on the traditional use of medicinal plants, its corresponding extracts, and even phytochemicals, in the treatment of OML, and attempts to be useful to scholars, scientists and health professionals working on drug discovery to develop effective anti-OML drugs. Numerous reports have assessed the efficacy of plant-derived bioactive molecules in the treatment of OML, commonly used in traditional Chinese, Ayurveda, Indian and Persian Medicine. Due its traditional uses, the efficacy of certain herbal preparations has been substantially investigated. Oral mucosal lesions have many triggering factors and plant-based treatments intervene through displaying diverse mechanisms of action, such as antioxidant, analgesic, anti-inflammatory, antifungal, antiseptic and anticarcinogenic effects. The in vitro and in vivo efficacy of certain herbal remedies and phytochemicals, including curcumin, lycopene, chamomile and *A. vera*, have been exhaustively described in malignant, or potentially malignant, oral diseases. In this sense, the studies reviewed here are the basis to discover new, effective and safer anti-oral mucosal lesion drugs, from a natural origin, which would be of great promise on OML management. More efficient clinical studies are therefore required for further validation, and further efforts are needed to characterize the active principles of medicinal plants in the treatment of OML.

## Figures and Tables

**Figure 1 biomolecules-09-00106-f001:**
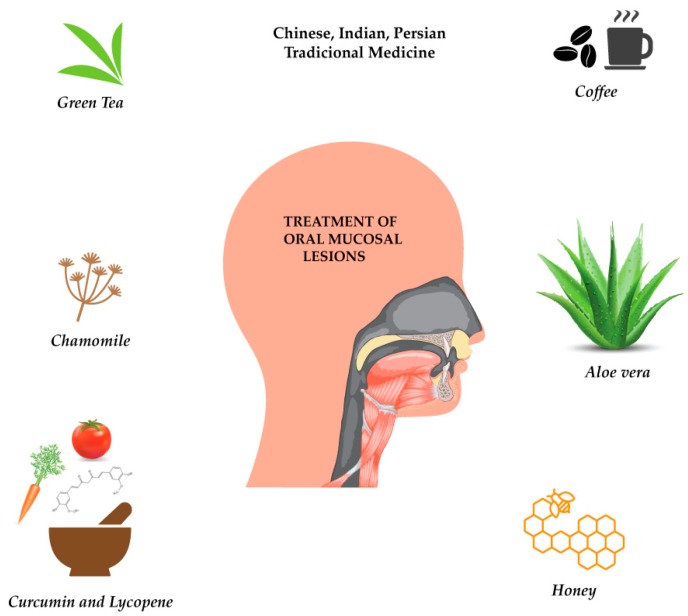
Plants and plant-derived bioactives in oral mucosal lesions.

**Figure 2 biomolecules-09-00106-f002:**
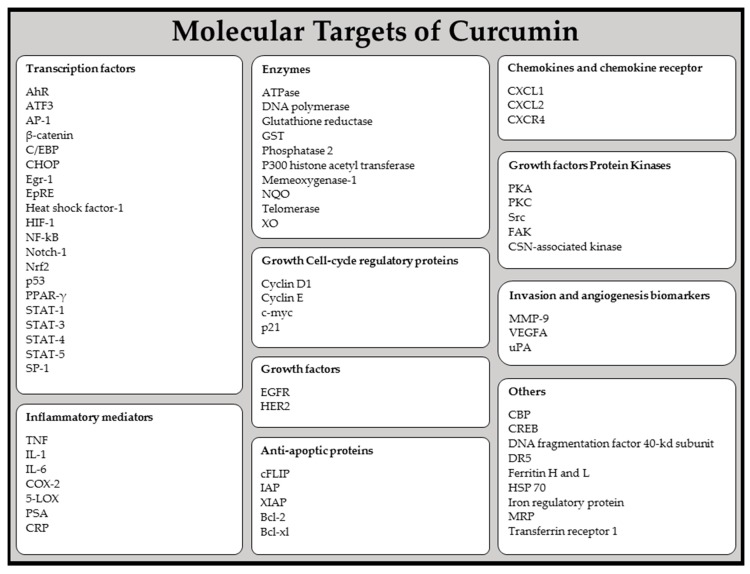
Molecular targets of curcumin. AhR, aryl hydrocarbon receptor; ATF, activating transcription factor; AP, activator protein; CBP, P300/CREB-binding protein; CHOP, C/EBP homologous protein; COP, constitutive photomorphogenic; COX, cyclooxygenase; CREB, cAMP response element binding; CSN, COP9 signalosome; CXCL, chemokine ligand; CXCR chemokine receptor; cFLIP, cellular FLICE-like inhibitory protein; CRP, C-reactive protein; DR, death receptor; EGFR, epidermal growth factor receptor; Egr, Early growth response; EpRE, electrophile response element; FAK, focal adhesion kinase; GST, glutathione *S*-transferase; HIF, hypoxia inducible factor; HSP, heat-shock protein; IAP, inhibitor of apoptosis protein; IL, interleukin; LOX, lipoxygenase; MMP, matrix metallopeptidase; MRP, multidrug resistance protein; NQO, naphthoquinone oxidoreductase; NADP(H), nicotinamide adenine dinucleotide phosphate (reduced form); Nrf2, NFE2-related factor; NF-κB, nuclear factor-kappa B; MRP, multi-drug resistance protein; PPAR-γ, peroxisome-proliferator-activated receptor-gamma; PKA, protein kinase A; PKC, protein kinase C; PSA, prostatic specific antigen; STAT, signal transducers and activators of transcription protein; TNF, tumor necrosis factor; uPA, urokinase plaminogen activator; VEGF, vascular endothelial growth factor; XIAP, X-linked IAP, XO, xanthine oxidase.

**Table 1 biomolecules-09-00106-t001:** Properties and signaling pathways of curcumin.

**Anti-inflammatory**	COX-1, COX-2, LOX, TNF-α, IFN-γ, iNOS and NF-κB inhibition [76] Down-regulates MCP-1 expression [77] Inhibits inflammatory cytokines production: IL-1β, IL-2, IL-5, IL-6, IL-8, IL-12, IL-18, MIP-1α [78] Down-regulates mitogen-activated and Janus kinases [79]
**Anti-neoplastic via cell-cycle arrest**	Cyclin D1 and CDK4 inhibition and p53, pRb, p21 and p27 up-regulation [80,81] Induced retinoblastoma protein [81] and STAT3 [82] phosphorylation and down-regulates cyclin D1 and cyclin E expression [81,82]
**Induction of apoptotic signals**	Induced up-regulation of Fas, FasL and DR5 expression [83] Enhances procaspases 3, 8 and 9 and poly(ADP-ribose) polymerase cleavage [84]

COX, cyclooxygenase; DR, death receptor; IFN, interferon; IL, interleukin; iNOS, inducible nitric oxide synthase; CDK, cyclin-dependent kinase; LOX, lipoxygenase; MCP, monocyte chemoattractant protein; MIP, macrophage inflammatory protein; NF-κB, nuclear factor kappa B; STAT, signal transducer and activator of transcription; TNF, tumor necrosis factor, pRB retinoblastoma protein; STAT, signal transducer and activator of transcription; FasL, Fas ligand.

**Table 2 biomolecules-09-00106-t002:** Actions of *Aloe vera* components.

Action	Treatment	Reference
*A. vera’s* glycoproteins fraction accelerates lesion healing and cell proliferation in rats	Ointment 10 mg/g/day for eight days, topically applied in an affected area	[129]
*A. vera* cream reduces hot water-induced lesion size in rat skin and increases re-epithelialization	A cream containing 0.5% *A. vera* in powder form applied 24 h after damage for 25 days	[130]
*A. vera* gel reduces inflammation and increases immunoglobulin E in a dermatitis mouse model	50 mg/kg/day for six weeks orally	[131]
*A. ferox* and *A. vera* prevent bacterial and fungal growth in rat and rabbit lesions. Preparations do not present secondary effects and help lesion healing	*A. vera* or *A. ferox* juice Rat model: 2 mL/8 h/2 days applied topically to lesions Rabbit model: 3 mL/6 h/4 days topically applied to lesions	[132]

**Table 3 biomolecules-09-00106-t003:** Human evidence of plants and phytochemicals against oral mucosal lesions.

Phytochemicals/Plants	Diseases	Observations	Ref.
Curcumin	Leukoplakia, lichen planus, and oral submucous fibrosis	Anti-tumor activity increasing vitamins C and E levels and preventing lipid peroxidation and DNA damage	Rai et al. [146]
	Oral lichen planus	Reduction of symptoms and signs	Chainani-Wu et al. [147]
		Reduction in signs (erythema and ulceration level)	Chainani-Wu et al. [148]
		Signs disappeared with no adverse effects	Sumanth et al. [149]
Curcumin, *Bidens pilosa*	Chemoradiotherapy-induced oral mucosal lesion	Prevention and treatment without associated inflammatory process	dos Santos et al. [150] Kashyap et al. [151]
Lycopene	Moderate periodontitis, moderate gingivitis	Effect as an adjunct to oral prophylaxis	Belludi et al. [152]
	Gingivitis	Reduced gingivitis, bleeding index and non-invasive measures of plaque	Chandra et al. [76]
Lycopene, β-carotene	Leukoplakia	Association between leukoplakia and low serum lycopene and β-carotene levels	Nagao et al. [153]
	Atrophic/erosive oral lichen planus	Association between oral lichen planus and low serum lycopene levels	Nagao et al. [154]
Green tea	Periodontal disease	Bactericidal effect	Hirasawa et al. [155]
	Chronic periodontitis	Green tea dentifrice improved probing depth, gingival index, and clinical attachment level	Hrishi et al. [156]
Coffee	Cancer of the oral cavity and pharynx	Caffeinated coffee intake was inversely related to a cancer risk of the oral cavity and pharynx	Galeone et al. [157]
Coffee, honey	Oral mucositis	Oral mucositis can be successfully treated by a combination of honey and coffee	Raeessi et al. [158]
Chamomile	Methotrexate-induced oral mucositis	Successfully reduced with mouthwash treatment	Mazokopakis et al. [159]
	Oral mucositis	Lower mouth pain score and fewer ulcerations	dos Reis et al. [160]
*Aloe vera*	Radiation-induced mucositis	No beneficial effect reported as an adjunct to head-and-neck radiotherapy	Su et al. [161]
	Oral candidiasis	Reduced oral candidiasis in patients with head and neck radiotherapy	Ahmadi et al. [162]
	Oral lichen planus	Reduced pain and burning sensation score, size and clinical characteristics of the lesions	Mansourian et al. [163]
		Induced clinical and symptom improvement	Choonhakarn et al. [164]

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
