# Peer review of "Plant-Derived Bioactives in Oral Mucosal Lesions: A Key Emphasis to Curcumin, Lycopene, Chamomile, Aloe vera, Green Tea and Coffee Properties"

_biomolecules, 2019, doi:10.3390/biom9030106_

Round 1

Reviewer 1 Report

The paper cannot be accepted for publication in its current form. 

It is not well structured and focused on the subject mentioned in title and abstract.

In the first part, the authors make an insufficient literature screening of plant species used in traditional Chinese, Indian and Persian medicine to treat different oral health problems.

Then, the authors present rather general data on the mechanisms of action of different chemicals, like curcumin and lycopene but, they include description of plant species, like Aloe vera, chamomile, tea and coffee in the category of phytochemicals??

However, chapter 4 has nothing to do with the declared purpose of this review.

Actually, the paper is more about the use of Aloe vera, chamomile, tea, coffee, lycopene and curcumin in oral medicine.

Therefore, the title and abstract should be adjusted to the real content of the paper, and however, the paper needs to be rewritten in a more focused way, with a lot of general information deleted or drastically shortened.

Author Response

The paper cannot be accepted for publication in its current form. It is not well structured and focused on the subject mentioned in title and abstract.

Answer: The Authors thank the reviewer for the important suggestions provided. We tried to address and answer the reviewer’s queries and hope to have clarified all points.

In the first part, the authors make an insufficient literature screening of plant species used in traditional Chinese, Indian and Persian medicine to treat different oral health problems.

Answer: The authors understand and agree with the Reviewer. In fact, the authors selected specific plants species and plant extracts. But the authors recognize that this information was lacking in the manuscript. Additionally, as this was also asked by Reviewer 2, the methodology is now indicated in the abstract, and the Reviewer can now kindly check the plant and plant-derived-phytochemicals selection made by the authors.

Then, the authors present rather general data on the mechanisms of action of different chemicals, like curcumin and lycopene but, they include description of plant species, like Aloe vera, chamomile, tea and coffee in the category of phytochemicals?? However, chapter 4 has nothing to do with the declared purpose of this review.

Answer: The Reviewer is completely correct. According to thisimportant suggestion, the abstract was amended, chapter 4 was changed and the manuscript adjusted and reorganized. Moreover, the manuscript was all revised before submission. 

Actually, the paper is more about the use of Aloe vera, chamomile, tea, coffee, lycopene and curcumin in oral medicine. Therefore, the title and abstract should be adjusted to the real content of the paper, and however, the paper needs to be rewritten in a more focused way, with a lot of general information deleted or drastically shortened.

Answer: The Reviewer is completely correct. According to thisimportant suggestion, the abstract was amended, chapter 4 was changed and the manuscript adjusted and reorganized. Moreover, the manuscript was all revised before submission. 

Reviewer 2 Report

The abstract should contain the methodology used for the review. The key words "oral mucosa" or "leukoplakia" or "oral lichen planus" are too much limited for the type of treatment told in the article (leucoplakia is not inherent to the use of medical plants etc). In the text is missing the division into articles of quality and quantity on the subject. In general, the work should reflect the publications so more relevant  through tables which reports topic and years of publications by other authors. Overall, the arguments treated in this article is innovative and could be considered after some corrections in every part of the text.

Author Response

The Authors thank the reviewer for the important suggestions provided. We tried to address and answer the reviewer’s queries and hope to have clarified all points.

The abstract should contain the methodology used for the review. 

Answer: The methodology is now indicated in the abstract, as kindly asked. 

The key words "oral mucosa" or "leukoplakia" or "oral lichen planus" are too much limited for the type of treatment told in the article (leucoplakia is not inherent to the use of medical plants etc). 

Answer: The authors understand the opinion of the Reviewer. In fact, several and very recent manuscripts have been studying the possibility of using a various number of plants for the treatment of these disorders: DOI:10.3390/ijms17091414, DOI:10.1016/j.mrrev.2017.09.001, DOI:10.1021/acs.chemrestox.7b00242, DOI: 10.1111/odi.12631, DOI:10.1111/jop.12083, DOI:10.1016/j.adaj.2017.02.053.

However, in order to make sure there would not be any mistakes, the authors fully reviewed the text and confirmed that the concepts were correct in their context.

In the text is missing the division into articles of quality and quantity on the subject. In general, the work should reflect the publications so more relevant through tables which reports topic and years of publications by other authors. Overall, the arguments treated in this article is innovative and could be considered after some corrections in every part of the text.

Answer: The authors understand the point of the Reviewer. However, the reason that led the authors to do a table with publications is that the manuscript has already several tables (6) and figures (4) and one more table with that much information would make the manuscript too heavy to read. Nevertheless, even if after this explanation, the reviewer still believes that this would me mandatory, the authors will do so.

Round 2

Reviewer 1 Report

The manuscript still need to be revised, the information more synthesized and reorganized. 

Author Response

The reviewer comment was carefully addressed and the manuscript revised accordingly.

Reviewer 2 Report

In the text is missing the division into articles of quality and quantity on the subject. In general, the work should reflect the publications so more relevant through tables which reports topic and years of publications by other authors. Overall, the arguments treated in this article is innovative and could be considered after some corrections in every part of the text.

Author Response

Thank you for the reviewer suggestion and appreciation of our work. We completely modified the overall manuscript, as also we clearly stated its main focus. The title was also changed in order to properly define the whole information present.

Round 3

Reviewer 1 Report

The authors have addressed the last suggestions. 

Reviewer 2 Report

In the table 6 should be inserted the name of the author of the study mentioned in a first column  and not the reference number.